# Improving Tail Label Prediction for Extreme Multi-label Learning

## Abstract

Extreme multi-label learning (XML) works to annotate objects with relevant labels from an extremely large label set. Many previous methods treat labels uniformly such that the learned model tends to perform better on head labels, while the performance is severely deteriorated for tail labels. However, it is often desirable to predict more tail labels in many real-world applications. To alleviate this problem, in this work, we show theoretical and experimental evidence for the inferior performance of representative XML methods on tail labels. Our finding is that the norm of label classifier weights typically follows a long-tailed distribution similar to the label frequency, which results in the over-suppression of tail labels. Base on this new finding, we present two new modules: (1) RANKNET learns to re-rank the predictions by optimizing a population-aware loss, which predicts tail labels with high rank; (2) TAUG augments tail labels via a decoupled learning scheme, which can yield more balanced classification boundary. We conduct experiments on commonly used XML benchmarks with hundreds of thousands of labels, showing that the proposed methods improve the performance of many state-of-the-art XML models by a considerable margin (6% performance gain with respect to PSP@1 on average).

## 1 Introduction

Extreme multi-label learning (XML) aims to annotate objects with relevant labels from an extremely large candidate label set. Recently, XML has demonstrated its broad applications. For example, in webpage categorization Partalas et al. (2015), millions of labels (categories) are collected in Wikipedia and one wishes to annotate new webpages with relevant labels from a huge candidate set; in recommender systems McAuley et al. (2015), one hopes to make informative personalized recommendations from millions of items. Because of the high dimensionality of label space, classic multi-label learning algorithms, such as Zhang & Zhou (2007); Tsoumakas & Vlahavas (2007), become infeasible. To this end, a number of computational efficient XML approaches are proposed Weston et al. (2011); Agrawal et al. (2013); Bi & Kwok (2013); Yu et al. (2014); Bhatia et al. (2015); E.-H. Yen et al. (2016); Yeh et al. (2017); Yen et al. (2017); Tagami (2017).

In XML, one important statistical characteristic is that labels follow a long-tailed distribution as illustrated in Figure 4 (left). Most labels occur only a few times in the dataset. Infrequently occurring labels (referred to as *tail label*) possess limited training samples and are harder to predict than frequently occurring ones (referred to as *head label*). Many existing XML approaches treat labels with equal importance, such as Prabhu & Varma (2014); Babbar & Schölkopf (2017); Khandagale et al. (2019), while Wei & Li (2018) demonstrates that most predictions of well-established methods are heads labels. However, in many real-world applications, it is still desirable to predict more tail labels which are more rewarding and informative, such as recommender systems Jain et al. (2016); Babbar & Schölkopf (2019); Wei & Li (2018); Wei et al. (2019).

To improve the performance for tail labels, existing solutions typically involve optimizing loss functions that are suitable for tail labels Jain et al. (2016); Babbar & Schölkopf (2019), leveraging the sparsity of tail labels in the annotated label matrix Xu et al. (2016), and transferring knowledge from data-rich head labels to data-scarce tail labels K. Dahiya (2019). These methods typically achieve better performance on tail labels than standard XML methods which treat labels equally, while they

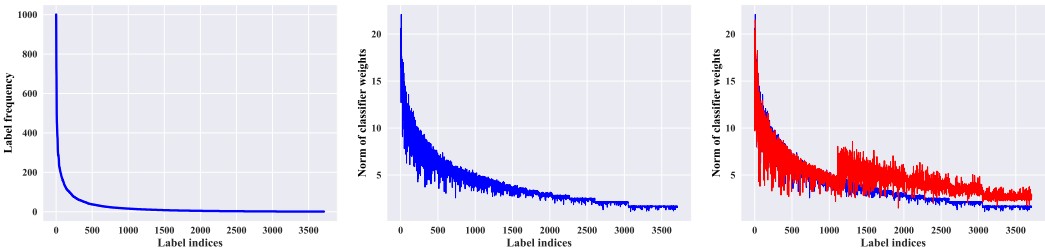

Figure 1: Left: Label frequency follows a long-tailed distribution. Middle: Norm of classifier weights of Bonsai models Khandagale et al. (2019). Right: Norm of classifier weights of Bonsai models when decoupled tail label augmentation is applied.

usually involve high computational costs. Moreover, previous studies do not explicitly explain the underlying cause of the inferior performance of many standard XML methods for tail labels.

In this work, we disclose theoretical and experimental evidence for the inferior performance of previous XML methods on tail labels. Our finding is that the norm of label classifier weights follows a long-tailed distribution similar to the label frequency as shown in Figure 4 (middle), and the prediction score of tail labels thereby is underrated. To alleviate this problem, we propose to rectify the classifier's outputs and training data distribution such that the prediction of tail labels is enhanced. We present two general modules suitable for any well-established XML methods: (1) RANKNET learns to re-rank the predictions by optimizing a population-aware loss function, which predicts tail labels with high rank; (2) TAUG augments tail labels via a decoupled learning scheme, which reduces the skewness of training data and yields more balanced classification boundary.

We conduct experiments to verify the effectiveness of the aforementioned instantiations. From our extensive studies across four benchmark datasets, we make the following intriguing contributions:

- We show that from both theoretical and experimental perspectives, the norm of label classifier weights follow a long-tailed distribution, i.e., the norms of head label classifier weights are considerably larger than that of tail label classifiers, which is a key cause of the inferior performance of many XML methods on tail labels.
- We propose two general modules: RANKNET for prediction score re-ranking by optimizing a new population-aware loss, and TAUG for decoupled tail label augmentation. Both methods can be paired with any XML model without changing the model.
- Experiments verify that our proposed modules achieve significant improvements (6% w.r.t. PSP@1 on average) for well-established XML methods on benchmark datasets.
- We provide an ablation study to highlight the effectiveness of each individual factor.

## 2 PREVIOUS EFFORTS

Existing work on XML can be roughly categorized as three directions:

**One-vs-all methods.** This branch of work trains classifiers for each label separately. Due to the huge size of label set, parallelization Babbar & Schölkopf (2017), label partitioning Khandagale et al. (2019), and label filter Niculescu-Mizil & Abbasnejad (2017) techniques are used to facilitate efficient training and testing. To alleviate memory overhead, recent works restrict the model capacity by imposing sparse constraints E.-H. Yen et al. (2016) or removing spurious parameters Babbar & Schölkopf (2017). One criticism of one-vs-all methods is that it fails to capture label correlations.

**Embedding-based methods.** Along this direction, researchers have proposed to embed the feature space and label space onto a joint low-dimensional space, then model the correlation between features and labels in hidden space Tai & Lin (2012); Chen & Lin (2012); Yu et al. (2014); Bhatia et al. (2015); Tagami (2017); Evron et al. (2018). This method can dramatically reduce the model parameters compared with the one-vs-all methods, but involves solving complex optimization problems.

**Tree-based methods.** In comparison to other types of approaches, tree-based methods greatly reduce inference time, which generally scales logarithmically in the number of labels. There are typically

two types of trees including instance trees Prabhu & Varma (2014); Siblini et al. (2018) and label trees Daume III et al. (2016); You et al. (2018), depending whether instance or label is partitioned in tree nodes. Tree-based methods usually suffer from low prediction accuracy affected by the cascading effect, where the prediction error at the top cannot be corrected at a lower level.

These methods can readily scale up to problems with hundreds of thousands of labels. However, Wei & Li (2018; 2019) claims that head labels make a significantly higher contribution to the performance than tail labels. Therefore, many work are conducted to improve the performance for tail labels.

**Optimization.** Jain et al. (2016) proposes propensity scored loss functions that promote the prediction of tail label with high ranks. Xu et al. (2016) decomposes the label matrix into a low-rank matrix and a sparse matrix. The low-rank matrix is expected to capture label correlations, and the sparse matrix is used to capture tail labels. Babbar & Schölkopf (2019) views tail label from an adversarial perspective and optimizes hamming loss to yield a robust model.

**Knowledge transfer.** K. Dahiya (2019) trains two deep models on head labels and tail labels. The semantic representations learned from head labels are transferred to the tail label model.

These methods achieve better performance on tail labels than standard XML methods which treat labels equally, while they do not explicitly explain the underlying cause of the inferior performance of many standard XML methods for tail labels. In this work, we find that the classification boundary of existing XML methods is skewed to head labels, causing the inferior performance.

## 3    METHODOLOGY

In XML, as we possess fewer data about tail labels, models learned on long-tailed datasets tend to exhibit inferior performance on tail labels Wei & Li (2018). However in practice, it is more informative and rewarding to accurately predict tail labels than head labels Jain et al. (2016). In this work, we attempt to alleviate this problem from the perspective of the classification boundary. We make an observation that the norm of label classifier weights follow a long-tailed distribution similar to the label frequency, which means that the prediction of tail labels is over-suppressed. This finding provides an evidence for us to improve the prediction of tail labels. We present ways of rectifying the classifier's outputs and data distribution via re-ranking and tail label augmentation, respectively.

**Notations.** We first describe notations used through the paper. Let $\mathbf{X} = \{\mathbf{x}_i\}_{i=1}^N$, $\mathbf{Y} = \{\mathbf{y}_i\}_{i=1}^N$ be a training set of size $N$, where $\mathbf{y}_i$ is the label vector for data point $\mathbf{x}_i$. Formally, XML is the task of learning a function $f$ that maps an input (or instance) $\mathbf{x} \in \mathbb{R}^D$ to its target $\mathbf{y} \in \{0,1\}^L$. We denote $n_j = \sum_{i=1}^N \mathbf{y}_{ij}$ as the frequency of the $j$-th label. Without loss of generality, we assume that the labels are sorted by cardinality in non-increasing order, i.e., if $j < k$, then $n_j \geq n_k$, where $1 \leq j, k \leq L$. In our setting, we have $n_1 \gg n_L$. According to the label frequency, we can split the label set into head labels and tail labels by a threshold $\tau \in (0, 1)$. We denote head label set $\mathcal{H} = \{1, \ldots, \lfloor \tau L \rfloor\}$ and tail label set $\mathcal{T} = \{\lfloor \tau L \rfloor + 1, \ldots, L\}$. $\tau$ is a user-specified parameter.

### 3.1    THE LONG-TAILED DISTRIBUTION OF CLASSIFIER WEIGHTS NORM

We present a different perspective regarding XML model, showing its inferior performance on tail labels is due to the imbalanced classification boundary. In Figure 4 (middle), we empirically observe that the norm of label classifier weights follows a similar long-tailed distribution as the label frequency. The results are produced on EUR-Lex dataset using a representative one-vs-all method Bonsai Khandagale et al. (2019). A similar observation on Wiki10-31K dataset is presented in the supplementary material. Since the norm of tail label classifier weights is considerably smaller than that of head label classifier weights, the predicted score of tail labels are typically underestimated in inference. We further support our finding theoretically and demonstrate the fact that the small norm of tail label classifier weights is the root cause of inferior performance.

We make the following mild assumption on the data: every input $\mathbf{x}$ is sampled from feature space completely at random, and there exists a constant threshold $t > 0$ for the input $\mathbf{x}$, such that the top-$k$ prediction for $\mathbf{x}$ is made as $\beta^{(k)} = \{y_l \mid \hat{\mathbb{P}}(y_l \mid \mathbf{x}) \geq t, 1 \leq l \leq L\}$, where $\hat{\mathbb{P}}(y_l \mid \mathbf{x})$ denotes the estimated label distribution. We assume $\mathbf{W} = \{\mathbf{w}_j\}_{j=1}^L$ be the weight matrix of a standard XML method. In particular, for binary relevance and tree-based classifier, $\mathbf{W}$ can be obtained by

optimizing Eq. (1), where $\mathcal{L}$ denotes the loss function, e.g., squared hinge loss, and constant $\lambda$ is a trade-off parameter. Note that for some tree-based methods, such as Bonsai Khandagale et al. (2019) and Parabel Prabhu et al. (2018), we consider $\mathbf{W}$ be the label classifier weights in leaf nodes, i.e., excluding meta-labels of internal tree nodes.

$$\min_{\mathbf{w}_j} \|\mathbf{w}_j\|_2^2 + \lambda \sum_{i=1}^{N} \mathcal{L}\left(\mathbf{Y}_{i,j}, \mathbf{w}_j^T \mathbf{x}_i\right), \forall 1 \leq j \leq L \tag{1}$$

For deep learning methods, we denote $\mathbf{W}$ be the weights of the last linear layer for classification by optimizing Eq. (2), where $\sigma$ is the softmax function, $f_\theta$ is the feature extractor parameterized by $\theta$, and $\mathcal{L}$ denotes the selected loss function, e.g., binary cross entropy. Note that this interpretation can also be adapted to typical embedding-based methods, such as Yu et al. (2014), where $f_\theta$ is linear and $\sigma$ is the identity function.

$$\min_{\mathbf{W}} \sum_{i=1}^{N} \mathcal{L}\left(\mathbf{y}_i, \sigma\left(\mathbf{W}^\top f_\theta\left(\mathbf{x}_i\right)\right)\right) \tag{2}$$

With the above setup, we summarize our findings in Theorem 1.

**Theorem 1.** *Let $\mathcal{D} = \{(\mathbf{x}_i, \mathbf{y}_i)\}_{i=1}^{N}$ be a sample set and $W$, which can be decomposed as $\{\mathbf{w}_j\}_{j=1}^{L}$, be the label classifier weights learned on $\mathcal{D}$ by optimizing Eq. (1) and Eq. (2). For an uniformly sampled point $\mathbf{x}$ which is i.i.d. with points in $\mathcal{D}$, we have $\|\mathbf{w}_j\| \propto \mathbb{E}\left[y_j \in \beta^{(k)}\right], \forall 1 \leq j \leq L$, where $\beta^{(k)}$ denotes the $k$ top-ranked indices of predicted labels in $\hat{\mathbb{P}}(\mathbf{y} \mid \mathbf{x})$.*

This theorem shows that the need for re-balancing the classifier weights to improve the performance on tail labels. Motivated by our finding, in the following we propose two new modules and discuss their effectiveness on tail labels. Proof of this theorem can be found in the supplementary material.

## 3.2 RANKNET: PREDICTION SCORE RE-RANKING NETWORKS

We introduce a novel re-ranking module motivated by our finding that the prediction score of tail labels are usually over-suppressed. Conventionally, the ranked list of predicted labels is determined by sorting the predicted score by a XML model $f$. Since the prediction score of tail labels are usually underrated as analyzed above, to alleviate this problem, we propose a re-ranking module to prioritize tail labels with higher score than head labels. More specifically, for a given sample $\mathbf{x}$, the probability of $l$-th label $y_l$ being relevant to $\mathbf{x}$ is specified by $\hat{y}_l = f(\mathbf{x})_l = \hat{\mathbb{P}}(y_l \mid \mathbf{x})$, which can be estimated by any existing XML model $f$. A RankNet block is a computational unit which maps the raw predictions to its enhanced population-aware predictions. Formally, given the raw prediction $\hat{\mathbf{y}} = f(\mathbf{x})$ and $\hat{\mathbf{y}} \in \mathbb{R}^{L \times 1}$, a RankNet building block is defined as:

$$F(\mathbf{x}) = \mathbf{W}^r \hat{\mathbf{y}} + \hat{\mathbf{y}},$$

where $\mathbf{W}^r \in \mathbb{R}^{L \times L}$ is learnable parameters of this RankNet block which is able to capture label correlations. To train RankNet, we propose a new population-aware loss function to minimize, which enforces higher predictions for tail labels than head labels and maintains prediction accuracy:

$$\mathcal{L}(F(\mathbf{x}), \mathbf{y}) = \sum_{i \in L_\mathbf{y}} \sum_{j \in L_\mathbf{y}} \left[\frac{n_i - n_j}{n_i + n_j}\left(F(\mathbf{x})_i - F(\mathbf{x})_j\right)\right]_+ + \lambda \sum_{i \in L_\mathbf{y}} \sum_{k \notin L_\mathbf{y}} [F(\mathbf{x})_k - F(\mathbf{x})_i + c]_+.$$

Here, $[z]_+ := \max(0, z)$ and $L_\mathbf{y}$ denotes the set of indices that are non-zero in $\mathbf{y}$, where $|L_\mathbf{y}| \ll L$. $c \geq 0$ is a constant which controls the margins. In particular, the first term enforces $F(\mathbf{x})_i < F(\mathbf{x})_j$ if $n_i > n_j$ and the denominator $n_i + n_j$ is used for normalization. The second term aims to retain the predictive accuracy by enforcing predicted score of relevant labels to be larger than that of irrelevant labels. Finally, $\lambda$ is used to balance these two terms. Note that since $|L_\mathbf{y}| \ll L$, the first term can be calculated effectively. To further speedup the training, a subset of irrelevant labels is sampled randomly. Therefore, RankNet introduces limited computational overhead, hence more expressive networks can be used. In other words, our design for function $F$ can be formally defined as:

$$F(\mathbf{x}) = \mathbf{W}^{r_2} \sigma(\mathbf{W}^{r_1} \hat{\mathbf{y}}) + \hat{\mathbf{y}},$$

where $\mathbf{W}^{r_1} \in \mathbb{R}^{V \times L}$ and $\mathbf{W}^{r_2} \in \mathbb{R}^{L \times V}$ are learnable weights using gradient descent, $V \ll L$ is the dimension of bottleneck layer to reduce the number of parameters, and $\sigma$ is the activation function.

Actually, we can stack any number of RankNet blocks to form a deep RankNet module and the output of each block is an enhancement over the output of the previous block. By doing so, the enhanced predictions $F(\mathbf{x})$ is able to aware the population of labels. We also provide a simple instantiations of $F(\mathbf{x})$ as $F(\mathbf{x})_l = r_l f(\mathbf{x})_l$ for the $l$-th label, where $r_l$ represents the inverse propensity score: $r_l = 1 + C (n_l + B)^{-A}$. $A, B, C$ are set as recommended in paper Jain et al. (2016). Finally, the top-ranked predictions in $F(\mathbf{x})$, rather than $f(\mathbf{x})$, are selected as final predictions. As one can expect, it increases the propensities of tail labels in inference, thereby more tail labels are shortlisted in final predictions. Unlike existing re-ranking methods Jain et al. (2016), RankNet is a general architecture and it can be concatenated after any standard XML methods.

### 3.3 TAUG: DECOUPLED LEARNING SCHEME AND TAIL LABEL AUGMENTATION

From another point of view, since the root cause of the imbalance of classifier norms, which is caused by the long-tailed data distribution, we propose to resolve this problem by reducing the skewness of training data.

**Decoupling the learning of head label and tail label.** We propose to decouple the learning of head labels and tail labels, instead of learning models jointly. This has two main benefits: (1) decoupled learning scheme helps prevent from modeling highly imbalanced data, i.e., the data distribution within head labels and tail labels are relatively less imbalanced; (2) head label model and tail label model can be trained in a parallel manner which reduces the training time. Recall that $\mathcal{H}$ and $\mathcal{T}$ denote head label set and tail label set, respectively. We split the training set $\mathcal{D} = \{(\mathbf{x}_i, \mathbf{y}_i)\}_{i=1}^{N}$ into two parts: $\mathcal{D}^h = \{(\mathbf{x}_i, \mathbf{y}_i) \mid y_{ij} = 1, \forall j \in \mathcal{H}\}$ and $\mathcal{D}^t = \{(\mathbf{x}_i, \mathbf{y}_i) \mid y_{ij} = 1, \forall j \in \mathcal{T}\}$. Models are then respectively learned on $\mathcal{D}^h$ and $\mathcal{D}^t$. In inference, the prediction score of models are integrated.

**Tail Label Augmentation.** To better explore the data distribution of tail labels, we consider two data augmentation techniques, *Input dropout* and *Input swap*.

1. *Input dropout:* For a selected keep probability $\rho \in [0, 1]$ and an input sample $\mathbf{x}$, it produces an augmented input $\mathbf{x}' = \mathbf{x} \odot \text{Bernoulli}(\rho, D)$, where $\odot$ denotes element-wise multiplication.
2. *Input swap:* For each instance $\mathbf{x}$, two activated features are randomly identified and their values are swapped. This procedure can repeat multiple times. Formally, for a pair of feature coordinates $i, j$, where $1 \le i, j \le D$, we swap their values $x_i$ and $x_j$.

Note that both data augmentation methods are label-invariant. In other words, for a given sample $(\mathbf{x}, \mathbf{y})$ and its augmented instance $\mathbf{x}'$, we take $\mathbf{y}' = \mathbf{y}$ as the corresponding label vector of $\mathbf{x}'$. Importantly, it is observed that there is a significant variation in the input features of tail labels from training set to test set Babbar & Schölkopf (2019), by generating more similar samples, it discourages the model from fitting spurious patterns in input features when training data is scarce and it also promotes the model to be robust to the corruption of the input features. The proposed decoupled learning scheme and tail label augmentation methods are observed to yield more balanced classification boundary as demonstrated in Figure 4 (right).

## 4 EXPERIMENTS

**Datasets.** We perform experiments on four XML datasets which are publicly available from the XML repository. Detailed statistics are summarized in Table 1, where $\bar{L}$ denotes average labels per sample and $\bar{N}$ denotes average sample per label.[1]

Table 1: Statistics of datasets.

| Dataset | # Train | # Features | # Labels | # Test | $\bar{L}$ | $\bar{N}$ |
|---|---|---|---|---|---|---|
| EUR-Lex | 15,539 | 5,000 | 3,993 | 3,809 | 5.31 | 25.73 |
| AmazonCat-13K | 1,186,239 | 203,882 | 13,330 | 306,782 | 5.04 | 448.57 |
| Wiki10-31K | 14,146 | 101,938 | 30,938 | 6,616 | 18.64 | 8.52 |
| Amazon-670K | 490,499 | 135,909 | 670,091 | 153025 | 3.9 | 5.4 |

---

[1]Datasets are available at the Extreme Classification Repository.

**Implementation.** Without further specification, we set the label splitting threshold $\tau = 0.1$ for EUR-Lex, and $\tau = 0.01$ for AmazonCat-13K, Wiki10-31K, and Amazon-670K. For tail label augmentation, we fix n_aug $= 4$, which means four auxiliary data points are generated for each sample. Recommended settings are used for all XML algorithms as specified in their paper.

**Evaluation Metrics.** We evaluate XML models on the test set and report results with respect to the commonly used evaluation metrics, i.e., P@$k$, nDCG@$k$, PSP@$k$, and PSnDCG@$k$ (PSN@$k$), where $k \in \{1, 3, 5\}$. Due to limited space, we elaborate the definitions in the supplementary material.

### 4.1 How Does the Prediction Score Re-ranking Affect the Results

We evaluate the effectiveness of the proposed score re-ranking method RANKNET. We run three popular XML algorithms, including FastXML Prabhu & Varma (2014), Bonsai Khandagale et al. (2019), and Parabel Prabhu et al. (2018) for comparisons.

From Table 2, it is effortless to observe that in all cases, three XML methods employing prediction score re-ranking achieve significantly higher PSP@$k$ and PSnDCG@$k$ compared with their baselines. In particular, FastXML respectively achieves as much as 5.81%, 8.4%, 0.94%, and 2.39% overall improvement on four datasets across PSP@$k$ and PSnDCG@$k$. In comparison, Bonsai outperforms its baseline by a larger margin, i.e., 5.31%, 7.49%, 5.84%, and 2.64% on four datasets, respectively. Similarly, Parabel achieves performance gains comparable to Bonsai, i.e., 5.31%, 7.51%, 5.26%, and 2.66%. This demonstrates that RANKNET provides an effective way to rectify the predictions for existing XML models, by which the predicted score of tail labels are indeed over-suppressed.

Table 2: Comparison between well-established XML methods with (w/) and without (w/o) RANKNET w.r.t. PSP@$k$ and PSnDCG@$k$ (PSN@$k$). The biggest improvements are in bold.

| Dataset | Method | w/o RANKNET | | | Improvement w/ RANKNET | | | |
|---|---|---|---|---|---|---|---|---|
| | | PSP@1 | PSP@3 | PSP@5 | PSP@1 | PSP@3 | PSP@5 | Avg. |
| EUR-Lex | FastXML | 26.29 | 33.57 | 37.80 | +7.78 | **+5.60** | **+3.94** | **+5.25** |
| | Bonsai | 36.91 | 44.89 | 49.46 | **+8.15** | +4.59 | +2.36 | +5.03 |
| | Parabel | 36.42 | 44.08 | 48.46 | +7.79 | +4.68 | +2.17 | +4.88 |
| AmazonCat-13K | FastXML | 48.06 | 59.25 | 66.70 | **+12.31** | **+7.62** | **+3.10** | **+7.68** |
| | Bonsai | 51.07 | 62.37 | 68.84 | +11.76 | +6.15 | +2.32 | +6.74 |
| | Parabel | 49.52 | 61.13 | 67.87 | +11.62 | +6.34 | +2.37 | **+6.78** |
| Wiki10-31K | FastXML | 9.76 | 10.31 | 10.64 | +0.95 | +0.90 | +0.98 | +0.94 |
| | Bonsai | 11.79 | 13.44 | 14.71 | **+8.17** | **+5.10** | +3.29 | **+5.52** |
| | Parabel | 11.68 | 12.72 | 13.69 | +6.52 | +4.71 | **+4.01** | +5.08 |
| Amazon-670K | FastXML | 18.69 | 21.87 | 24.44 | +3.44 | **+2.49** | **+1.63** | **+2.52** |
| | Bonsai | 27.10 | 30.69 | 33.91 | **+3.84** | +2.45 | +1.12 | +2.47 |
| | Parabel | 26.35 | 29.94 | 33.16 | +3.74 | +2.48 | +1.27 | +2.50 |
| Dataset | Method | PSN@1 | PSN@3 | PSN@5 | PSN@1 | PSN@3 | PSN@5 | Avg. |
| EUR-Lex | FastXML | 26.29 | 31.59 | 34.40 | +7.78 | **+6.23** | **+5.14** | **+6.38** |
| | Bonsai | 36.91 | 42.43 | 45.23 | **+8.15** | +4.91 | +3.74 | +5.60 |
| | Parabel | 36.42 | 41.99 | 44.90 | +7.79 | +5.53 | +3.90 | +5.74 |
| AmazonCat-13K | FastXML | 48.06 | 56.13 | 61.00 | **+12.31** | **+8.99** | **+6.09** | **+9.13** |
| | Bonsai | 51.07 | 59.25 | 63.53 | +11.76 | +7.73 | +5.23 | +8.24 |
| | Parabel | 49.52 | 57.92 | 62.38 | +11.62 | +7.84 | +5.26 | +8.24 |
| Wiki10-31K | FastXML | 9.76 | 10.17 | 10.41 | +0.95 | 0.91 | +0.96 | +0.94 |
| | Bonsai | 11.79 | 13.03 | 13.92 | **+8.17** | **+5.85** | +4.54 | **+6.17** |
| | Parabel | 11.68 | 12.47 | 13.13 | +6.52 | +5.15 | **+4.62** | +5.43 |
| Amazon-670K | FastXML | 18.69 | 22.05 | 23.58 | +3.44 | +1.81 | +1.52 | +2.26 |
| | Bonsai | 27.10 | 29.74 | 31.94 | **+3.84** | **+2.83** | +1.92 | **+2.86** |
| | Parabel | 26.35 | 29.01 | 31.20 | +3.74 | +2.81 | **+1.99** | +2.85 |

## 4.2 How Does the Decoupled Tail Label Augmentation Affect the Results

In the following, we verify the effectiveness of the decoupled tail label augmentation. We choose Bonsai Khandagale et al. (2019) as our base model for its appealing performance as shown in Table 2.

*Bag-of-Words (BOW) vs. Dense Embedding.* Since many benchmark datasets for XML are text data, we find that the dense embedding used in Chang et al. (2020) achieves significant gains over BOW. We compare the results on EUR-Lex, and find that dense embedding respectively achieves 2.87% and 3.13% higher performance w.r.t. PSP@$k$ and PSnDCG@$k$ on average. We conduct experiments using dense embedding in the rest of this paper except for Amazon-670K, which is not available.

*Classifier Weights Normalization.* As a straightforward way of balancing the norm of classifier weights Kang et al. (2020), we examine the effectiveness of weights normalization. It does not show significant effect on the performance. In particular, it improves the performance by 0.63% w.r.t. PSnDCG@$k$, but drops the performance with 0.57% w.r.t. PSP@$k$, on EUR-Lex. This suggests that weights normalization, which equalizes the propensity of labels, is undesirable for XML.

*Tail Label Augmentation.* To justify our claim that it is beneficial to decouple the learning of head labels and tail labels, and augment the tail label, which generate more balanced classification boundaries. The results are reported in Table 3. Since the focus of this paper is the performance improvement for tail label, we report and compare the results in terms of PSP@$k$ and PSnDCG@$k$. As we can see from the results, TAUG achieves averagely 3%, 1.06%, 1.59%, 7.42% improvement w.r.t. PSP@$k$, and 3.5%, 1.08%, 1.41%, 7.54% w.r.t. PSnDCG@$k$, on four datasets. This demonstrates that the investigated two data augmentation techniques via decoupled learning scheme can help the learning of tail labels, by yielding more balanced classification boundary which predicts tail labels with relatively larger score compared with the baseline.

Table 3: Comparison between methods with (w/) and without (w/o) TAUG w.r.t. PSP@$k$ and PSnDCG@$k$ (PSN@$k$). The biggest improvements across four datasets are in bold.

| Dataset | w/o TAUG | | | Improvement w/ TAUG | | | |
|---|---|---|---|---|---|---|---|
| | PSP@1 | PSP@3 | PSP@5 | PSP@1 | PSP@3 | PSP@5 | Avg. |
| EUR-Lex | 39.66 | 47.92 | 52.28 | +4.64 | +2.69 | +1.67 | +3.00 |
| AmazonCat-13K | 49.41 | 60.39 | 66.49 | +2.22 | +0.73 | -0.24 | +1.06 |
| Wiki10-31K | 12.41 | 14.13 | 15.52 | +0.75 | +1.91 | +2.12 | +1.59 |
| Amazon-670K | 27.10 | 30.69 | 33.91 | **+10.21** | **+7.18** | **+4.87** | **+7.42** |
| Dataset | PSN@1 | PSN@3 | PSN@5 | PSN@1 | PSN@3 | PSN@5 | Avg. |
| EUR-Lex | 39.66 | 45.81 | 48.48 | +4.64 | +3.14 | +2.72 | +3.50 |
| AmazonCat-13K | 49.41 | 57.36 | 61.41 | +2.22 | +0.75 | +0.27 | +1.08 |
| Wiki10-31K | 12.41 | 13.69 | 14.68 | +0.75 | +1.65 | +1.83 | +1.41 |
| Amazon-670K | 27.10 | 29.74 | 31.94 | **+10.21** | **+7.99** | **+4.43** | **+7.54** |

## 4.3 Comparison with the State of the Arts

In Table 4, we compare the performance of the proposed modules with recent XML methods, including PfastreXML Jain et al. (2016), ProXML Babbar & Schölkopf (2019), AttentionXML You et al. (2018), and GLas Guo et al. (2019), that report state-of-the-art results on tail labels. Since PSnDCG@$k$ is unavailable for AttentionXML and GLaS, we report and compare the results w.r.t. PSP@$k$ in this part. We apply the proposed prediction score re-ranking module (RANKNET) individually and jointly with the decoupled tail label augmentation module (TAUG) for comparison. Though both AttentionXML and GLaS are carefully designed deep learning methods, it is surprising to see that our Bonsai based variants achieve the best results in 8 out of 12 cases, and the second-best results in other cases. Apart from deep learning methods, PfastreXML and ProXML are two leading approaches which achieve good performance on tail labels, while they are outperformed by our methods in most cases. In comparison with the baselines in Table 2, RANKNET+TAUG demonstrates more than 6% performance gains w.r.t. PSP@1 on average.[2]

---

[2]Our anonymous code is available in the supplementary material.

Table 4: Comparison with state-of-the-art methods w.r.t. PSP@$k$. Bold numbers are the best and underlined numbers are the second-best. RANKNET and RANKNET+TAUG are our proposed methods.

| Dataset | Metric | PfastreXML | ProXML | GLaS | AttentionXML | RANKNET | RANKNET+TAUG |
|---|---|---|---|---|---|---|---|
| EUR-Lex | PSP@1 | 43.53 | 45.20 | 49.77 | 44.97 | 48.44 | **51.86** |
|  | PSP@3 | 45.38 | 48.50 | 51.05 | 51.91 | 52.94 | **54.29** |
|  | PSP@5 | 47.02 | 51.00 | 53.82 | 54.86 | **55.69** | 55.45 |
| AmazonCat-13K | PSP@1 | **63.51** | 61.92 | 47.53 | 53.76 | 62.83 | 63.45 |
|  | PSP@3 | 68.71 | 66.93 | 62.74 | 68.72 | 68.52 | **69.96** |
|  | PSP@5 | 71.21 | 68.36 | 71.66 | **76.38** | 71.17 | 70.88 |
| Wiki10-31K | PSP@1 | 18.75 | 17.17 | - | 15.57 | 20.16 | **23.06** |
|  | PSP@3 | 18.47 | 16.07 | - | 16.80 | 19.65 | **21.60** |
|  | PSP@5 | 18.50 | 16.38 | - | 17.82 | 20.48 | **21.66** |
| Amazon-670K | PSP@1 | 29.28 | 30.31 | 38.94 | 30.29 | 30.94 | **40.41** |
|  | PSP@3 | 30.79 | 32.31 | **39.72** | 33.85 | 33.14 | 39.63 |
|  | PSP@5 | 32.40 | 34.43 | **41.24** | 37.13 | 35.04 | 39.54 |

## 4.4 HOW DOES THE STRENGTH OF DATA AUGMENTATION MATTER

We are interested in how would the strength of data augmentation affects the classifier weights and the model performance. In Figure 2 (left), we illustrate the norm of classifier weights by choosing n_aug $\in \{0, 1, 4, 8\}$, where n_aug indicates the number of augmented samples to generate for each data point. Note that when n_aug = 0, models are learned on the initial training data with augmentation. It can be noted that the norms of the tail label classifiers become larger as n_aug increases. In Figure 2 (right), the performance tends to improve as n_aug increases w.r.t. PSP@$k$. As one can expect, P@$k$ drops with a narrow margin. These results suggest that data augmentation can help re-balance the norm of classifier weights, which is beneficial to tail labels. In addition, we conduct ablation studies to compare two data augmentation techniques, i.e., *input dropout* and *input swap*, in the supplementary material. We also demonstrate the effect of different label splitting threshold $\tau$.

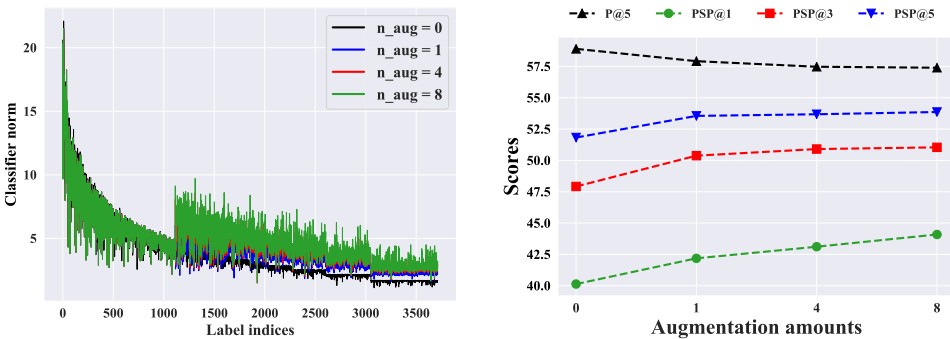

Figure 2: Left: Norms of classifier weights with varying n_aug. Right: The performance w.r.t. P@5 and PSP@$k$ as a function of n_aug. Results in both figures are produced using Bonsai on EUR-Lex.

## 5 CONCLUSION

In this paper, we show that from both theoretical and empirical perspectives, norm of label classifier weights follows long-tailed distribution, if labels are treated uniformly, which is a key cause of the inferior performance for tail labels. To alleviate this problem, we explore the re-ranking module that optimizes a new population-aware loss, and tail label augmentation module that decouples head labels and tail labels. Through extensive studies, our proposed two modules achieve significant performance gains. Moreover, both modules can be readily applied to any well-established XML methods without changing their models. We believe that our findings not only contribute to a deeper understanding of the tail label problem, but can offer inspiration for future work.

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

## A    Theoretical Justification of the Inferior Performance on Tail Labels

**Theorem 1.** *Let $\mathcal{D} = \{(\mathbf{x}_i, \mathbf{y}_i)\}_{i=1}^{N}$ be a sample set and $\mathbf{W}$, which can be decomposed as $\{\mathbf{w}_j\}_{j=1}^{L}$, be the label classifier weights learned on $\mathcal{D}$ by optimizing Eq. (1) and Eq. (2). For an uniformly sampled point $\mathbf{x}$ which is i.i.d. with points in $\mathcal{D}$, we have $||\mathbf{w}_j|| \propto \mathbb{E}\left[y_j \in \beta^{(k)}\right], \forall 1 \leq j \leq L$, where $\beta^{(k)}$ denotes the $k$ top-ranked indices of predicted labels in $\hat{\mathbb{P}}(\mathbf{y} \mid \mathbf{x})$.*

*Proof.* Without loss of generality, we assume $||\mathbf{w}_1|| \geq ||\mathbf{w}_2|| \cdots \geq ||\mathbf{w}_L|| > 0$. For any input $\mathbf{x}$, its prediction score is computed as $\hat{\mathbb{P}}(y_j \mid \mathbf{x}) = g(\mathbf{w}_j^\top \mathbf{x})$ for the $j$-th label, where $g(\cdot)$ is a monotonically increasing link function, such as the exponential function, the largest top-$k$ prediction score will be selected as the final predictions. For simplicity, we assume $g(z) = z$ as an identical function and our analysis can be easily extended to the exponential function. Suppose that $t \in (0, 1)$ is the threshold of input $\mathbf{x}$, such that the final prediction is $\beta^{(k)} = \{y_j \mid \mathbf{w}_j^\top \mathbf{x} \geq t, 1 \leq j \leq L\}$, where $|\beta^{(k)}| = k$. Here we assume there exists a small constant $k \ll L$ such that $\mathbf{w}_j^\top \mathbf{x} \geq 0, \forall 1 \leq j \leq k$, which is reasonable in extreme classification. Since $\mathbf{w}_j^\top \mathbf{x} = ||\mathbf{w}_j|| \cdot ||\mathbf{x}|| \cos \theta_j$, where $\theta_j$ denotes the included angle of classifier $\mathbf{w}_j$ and sample $\mathbf{x}$. Note that $\mathbf{x}$ is usually normalized in advance and $||\mathbf{x}||$ can be considered as a constant for different samples. In other words, the prediction can be rewritten as $\beta^{(k)} = \{y_j \mid \cos \theta_j \geq \frac{t}{||\mathbf{w}_j|| \cdot ||\mathbf{x}||}, 1 \leq j \leq L\}$. This can be considered as that $\mathbf{x}$ is sampled from a ball with radius equals $||\mathbf{x}||$ in the feature space completely at random, which means that $\theta_j$ is uniformly sampled from $[0, \pi]$. Note that we always have $||\mathbf{w}_j|| \geq 1, \forall 1 \leq j \leq L$ because the bias term of each label classifier is set to be 1, which means $\frac{t}{||\mathbf{w}_j||} < 1$. Let $b = \arccos \frac{t}{||\mathbf{w}_j|| \cdot ||\mathbf{x}||}$, we have $\mathbb{P}(y_j \in \beta^{(k)}) = \frac{b}{\pi}$. By taking the expectation over $\theta_j$, we have

$$\mathbb{E}\left[y_j \in \beta^{(k)}\right] = \int_0^\pi \mathbb{P}(y_j \in \beta^{(k)}) \, d\theta_j = b.$$

Since $b$ typically scales as $||\mathbf{w}_j||$, we conclude that the probability of the $j$-th label is included in top-$k$ predictions of input $\mathbf{x}$, is proportional to its classifier's norm $||\mathbf{w}_j||$, or formally

$$||\mathbf{w}_j|| \propto \mathbb{E}\left[y_j \in \beta^{(k)}\right], \forall 1 \leq j \leq L.$$

We empirically illustrate this finding in Figure 3. $\qquad\square$

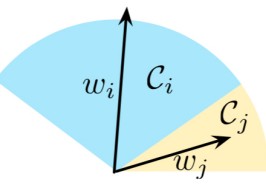

Figure 3: Illustration on different classifiers and their corresponding decision boundaries, where $\mathbf{w}_i$ and $\mathbf{w}_j$, $||\mathbf{w}_i|| > ||\mathbf{w}_j||$, denote the classification weights for class $i$ and $j$ respectively. $\mathcal{C}_i$ and $\mathcal{C}_j$ are the classification cone belongs to the $i$-th and $j$-th class in the feature space, respectively. The classifier with larger weights norm, i.e., $\mathbf{w}_i$, has wider decision boundary.

## B    Additional Experimental Results

### B.1    Virtualization of the Norms of Classifier Weights on Wiki10-31K

### B.2    Evaluation Metrics

*P@k.* Top-$k$ precision is a commonly used ranking based performance measure in XML and has been widely adopted for ranking tasks. In Top-$k$ precision, only a few top predictions of an instance

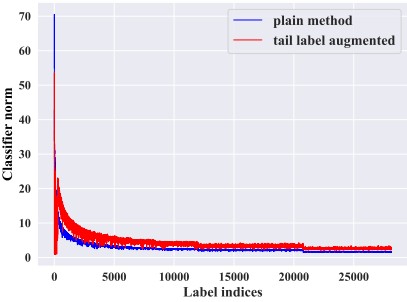

Figure 4: The norm of classifier weights of Bonsai models before (blue) and after (red) applying data augmentation for tail label. Results are produced on Wiki10-31K dataset.

will be considered. For each instance $\mathbf{x}$, the Top-$k$ precision is defined for a predicted score vector $\hat{\mathbf{y}} \in \mathcal{R}^L$ and ground truth label vector $\mathbf{y} \in \{-1, 1\}^L$ as

$$P@k := \frac{1}{k} \sum_{l \in \mathrm{rank}_k(\hat{\mathbf{y}})} \mathbf{y}_l,$$

where $\mathrm{rank}_k(\hat{\mathbf{y}})$ returns the indices of $k$ largest value in $\hat{\mathbf{y}}$ ranked in descending order.

*nDCG@$k$.* nDCG@$k$ is another commonly used ranking based performance measure:

$$\mathrm{nDCG}@k := \frac{\mathrm{DCG}@k}{\sum_{l=1}^{\min(k,\|\mathbf{y}\|_0)} \frac{1}{\log(l+1)}},$$

where $\mathrm{DCG}@k := \sum_{l \in \mathrm{rank}_k(\hat{\mathbf{y}})} \frac{\mathbf{y}_l}{\log(l+1)}$ and $\|\mathbf{y}\|_0$ returns the 0-norm of the true-label vector.

*PSP@$k$.* Propensity scored variants of such losses, including precision@k and nDCG@k, are developed and proved to give unbiased estimates of the true loss function even when ground-truth labels go missing under arbitrary probabilistic label noise models.

$$\mathrm{PSP}@k := \frac{1}{k} \sum_{l \in \mathrm{rank}_k(\hat{\mathbf{y}})} \frac{\mathbf{y}_l}{p_l}.$$

Here, $p_l$ is the propensity score for label $l$ which helps in making metrics unbiased.

*PSnDCG@$k$.* Similar to nDCG@$k$, its propensity scored variant is defined as

$$\mathrm{PSnDCG}@k := \frac{\mathrm{PSDCG}@k}{\sum_{l=1}^{k} \frac{1}{\log(l+1)}},$$

where

$$\mathrm{PSDCG}@k := \sum_{l \in \mathrm{rank}_k(\hat{\mathbf{y}})} \frac{\mathbf{y}_l}{p_l \log(l+1)}$$

### B.3 RESULTS OF RE-RANKING W.R.T. P@$k$ AND NDCG@$k$

As shown in Table 5, performance with respect to P@$k$ and nDCG@$k$ usually deteriorates. By using RANKNET, more tail labels are predicted with higher confidence than head labels, which would introduce more false-positive predictions. This shows that there is a trade-off between propensity scored measures and vanilla measures according to specific demands in applications.

### B.4 HOW DOES THE LABEL SPLITTING THRESHOLD MATTER

In Figure 5, we demonstrate how the splitting threshold affects the results. We experiment with $L_h = \lfloor \tau L \rfloor$ and $L_t = L - \lfloor \tau L \rfloor$, where $\tau \in \{0.1, 0.2, 0.3, 0.4, 0.5, 1\}$ for EUR-Lex and $\tau \in \{0.01, 0.02, 0.03, 0.04, 0.05, 1\}$ for Wiki10-31K. Note that, when $\tau = 1$, all labels are considered as the head label and no data augmentation is conducted. As $L_t$ decreases (i.e., $\tau$ increases), performance in terms PSP@$k$ typically drops slightly suggesting most labels are should be considered as the tail label and in the need for data augmentation.

Table 5: Comparison between methods with and without re-ranking in terms of P@$k$ and nDCG@$k$.

| Dataset | Method | P@1 | P@3 | P@5 | nDCG@3 | nDCG@5 |
|---|---|---|---|---|---|---|
| EUR-Lex | FastXML | 70.64 | 59.20 | 49.41 | 62.23 | 57.16 |
| | RANKNET$_{FastXML}$ | 72.40 | 60.26 | 50.20 | 63.42 | 58.20 |
| | Bonsai | 82.75 | 69.44 | 58.25 | 72.84 | 67.26 |
| | RANKNET$_{Bonsai}$ | 81.46 | 68.53 | 56.79 | 71.80 | 65.87 |
| | Parabel | 82.20 | 68.70 | 57.53 | 72.16 | 66.53 |
| | RANKNET$_{Parabel}$ | 80.67 | 68.34 | 56.29 | 71.56 | 65.34 |
| AmazonCat-13K | FastXML | 92.78 | 77.38 | 62.25 | 86.37 | 84.12 |
| | RANKNET$_{FastXML}$ | 90.69 | 76.75 | 62.38 | 85.29 | 83.60 |
| | Bonsai | 91.59 | 76.92 | 62.21 | 85.56 | 83.54 |
| | RANKNET$_{Bonsai}$ | 88.47 | 75.85 | 61.89 | 83.84 | 82.35 |
| | Parabel | 92.06 | 76.81 | 62.05 | 85.62 | 83.54 |
| | RANKNET$_{Parabel}$ | 88.23 | 75.83 | 61.50 | 83.81 | 82.03 |
| Wiki10-31K | FastXML | 82.93 | 68.01 | 57.98 | 71.41 | 63.58 |
| | RANKNET$_{FastXML}$ | 84.23 | 69.46 | 59.21 | 72.84 | 64.85 |
| | Bonsai | 84.58 | 73.72 | 64.44 | 76.25 | 69.19 |
| | RANKNET$_{Bonsai}$ | 82.58 | 71.86 | 63.67 | 74.31 | 68.02 |
| | Parabel | 84.18 | 72.46 | 63.37 | 75.21 | 68.21 |
| | RANKNET$_{Parabel}$ | 83.26 | 71.33 | 62.82 | 74.08 | 67.50 |
| Amazon-670K | FastXML | 35.89 | 31.80 | 28.58 | 34.03 | 32.79 |
| | RANKNET$_{FastXML}$ | 34.13 | 31.18 | 28.47 | 33.20 | 32.39 |
| | Bonsai | 45.48 | 40.22 | 36.37 | 42.64 | 40.85 |
| | RANKNET$_{Bonsai}$ | 43.96 | 39.32 | 35.77 | 41.66 | 40.11 |
| | Parabel | 44.91 | 39.76 | 35.98 | 42.11 | 40.33 |
| | RANKNET$_{Parabel}$ | 43.85 | 39.19 | 35.57 | 41.48 | 39.85 |

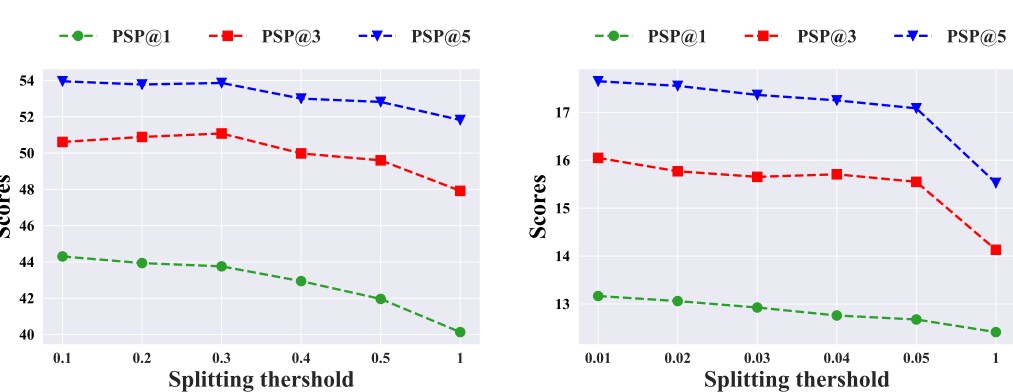

Figure 5: The performance in terms of PSP@$k$ as a function of splitting thresholds on EUR-Lex (left) and Wiki10-31K (right). Results are produced using Bonsai method.

### B.5 ABLATIONS ON THE INPUT DROPOUT AND INPUT SWAP

We conduct ablation studies to compare the effectiveness of the proposed two tail label data augmentation strategies. We compare four methods using Bonsai as the base model:

- *baseline*: this method does not use any data augmentation techniques.
- TAUG-*d*: this method uses *input dropout* only with n_aug = 4.

- TAUG-*s*: this method uses *input swap* only with n_aug = 4.
- TAUG-*ds*: this method uses both *input dropout* and *input swap* with n_aug = 4.

We report the results in Table 6. On EUR-Lex dataset, it is effortless to see that both input dropout and input swap achieve performance gains, i.e., respectively 2.67% and 2.06% on average, against the baseline method. It further improves the performance when both techniques are incorporated into the Bonsai, i.e., TAUG-ds. On Wiki10-31K dataset, it shows a relatively smaller margin of improvement. Nevertheless, TAUG-ds still improves the performance with 1.65% on average.

Table 6: Comparison between input dropout (-d) and input swap (-s) using dense embedding representations on EUR-Lex and Wiki10-31K datasets. The best results are in bold.

| Dataset | Method | PSP@1 | PSP@3 | PSP@5 | PSnDCG@3 | PSnDCG@5 |
|---------|--------|-------|-------|-------|----------|----------|
| EUR-Lex | baseline | 40.10 | 47.91 | 51.85 | 45.81 | 48.48 |
| | TAUG-d | 43.52 | 50.45 | 53.94 | 48.63 | 50.99 |
| | TAUG-s | 42.81 | 49.94 | 53.27 | 48.09 | 50.35 |
| | TAUG-ds | 44.20 | 50.58 | 53.97 | 48.91 | 51.19 |
| Wiki10-31K | baseline | 12.41 | 14.13 | 15.52 | 13.69 | 14.68 |
| | TAUG-d | 13.14 | 15.94 | 17.60 | 15.26 | 16.46 |
| | TAUG-s | 12.99 | 15.70 | 17.26 | 15.03 | 16.17 |
| | TAUG-ds | 13.18 | 16.05 | 17.63 | 15.34 | 16.50 |

