# OpenReview forum: "Improving Tail Label Prediction for Extreme Multi-label Learning"
_ICLR.cc/2021/Conference — Reject_

### Official Review · AnonReviewer4 · 2020-10-26
**A paper with some important flaws**

**Rating:** 3
**Confidence:** 4

**Review:**

This paper considers the setting of extreme multi-label classification, where labels typically follow a power-law distribution with many infrequently-observed labels (so-called tail labels). In this setting it often happens that multi-label classifiers more often predict frequent labels as positive than infrequent labels. In practical applications this is not always wanted, and the authors present a new algorithm that favors tail labels over frequent labels. To this end, a specific ranking-based loss function that consists of two parts is minimized. The first part of the loss ranks positive tail labels higher than positive frequent labels. The second part is more standard, and ranks positive labels higher than negative labels.

Improving predictions for tail labels is an interesting research goal that has not been thoroughly addressed in the literature, but I am not convinced of the theoretical results and the introduced algorithm.

Theorem 1 does not hold because an important condition is missing. The theorem would only hold if w_j^T x > 0 for all x. However, in practice, such a condition cannot be guaranteed. The formulation of the theorem is more difficult than needed, but what the authors want to say is the following: "P(y_j|x) is a monotonically increasing function of the norm of w_j". The proof that is found in the appendix cannot be correct because one can easily construct a counterexample when w_j^T x < 0, and the proof is also more complicated than needed. In fact P(y_j|x) is just a transformation of w_j^T x via a monotone function g with [0,1] as codomain. Useful choices for g are the logit or probit link, but not an exponential function (as stated in the proof). With this insight one can easily see that, when w_j^T x < 0, the probability P(y_j|x) will decrease when for example all coefficient in w are multiplied with a factor two. In that case the norm of w_j all increases, and we have a counterexample for the theorem. To my opinion, the proof makes a few very strange constructions, but I cannot immediately see where the mistake is.

I also do not understand why the link function is only introduced in the appendix, because it is a key concept to link w_j^T x and P(y_j|x). To increase readability, I would advise to discuss this early in Section 2. I also do not understand what w_j represents in the case of tree-based models. More discussion is needed. For tree-based models, one doesn't have a weight vector per class, isn't it?

I am also not convinced of the algorithm that is introduced in Section 3.2. The method is very ad-hoc, without any theoretical justification. As a result of pairwise terms, it might also be computationally challenging to optimize the proposed loss for extreme multi-label datasets. Isn't there a much simpler solution? Using the terminology of Section 2.1, one could simply improve the performance for tail labels by adjusting the threshold t for such labels only. Has such a simple solution been considered in literature? In that way one could fit standard probabilistic classifiers during training, following by a reasoning on probabilities in a post-training procedure. Similar to the approach of the authors, one could take label frequencies into account during this post-training procedure, resulting in a threshold t that depends on label frequency.

In the experiments it is not clear to me why only four XML datasets are used. Why were the other datasets in the XML repository not analyzed? Please provide a good motivation or analyze all datasets.

---

> ### Author Response · Authors · 2020-11-13
> **Response to Reviewer 3**
>
> Thank you for your comments. We address your concerns in the following point by ponit.
>
> 1. In term of Theroem 1, we did miss a condition. In specific, we need to make a assumption that there exists a small constant $k$ such that $w_j^T x \geq 0, \forall 1 \leq j \leq k$ for all x. Note that in XML, the number of labels $L$ is typically large, e.g., $L > 10^3$ and $k$ can be very small, e.g., we usually need to set $k =. 5$.
>
> 2. For tree-based methods, we do need to make the statement clear that only part of tree-based methods are applicable to the proposed framework, such as Bonsai and Parabel. However, for FastXML, one doesn't have a weight vector per class.
>
> 3. For better ways of improving performance on tail labels, adjusting the threshold for tails labels only has not been considered in the literature. Note that, in XML, only top $k$ labels are usually predicted for each testing points because the output label size is very large than conventional multi-label learning and it is not affordable to obtain prediction scores over all labels.
>
> 4. In terms of the used datasets, we conducted experiments on Amazon-670K which is the largest dataset we can run on our university computers. We believe the proposal can achieve similar performance on other datasets.

---

### Official Review · AnonReviewer2 · 2020-10-28
**Impressive results but weak baselines + Several key model details missing**

**Rating:** 5
**Confidence:** 4

**Review:**

Summary:
=======
In prediction problems with millions of labels also known as Extreme Multi-label Learning (XML) problems, e.g., recommender systems, the model predictions are not as good for the tail (rarer) labels. This paper proposes two models for this problem. The first model is re-ranking-based, that is, it reranks the prediction scores of a standard XML model. The second model tries to augment the rarer labels to reduce the skew in data. Results shown on several real-world datasets highlight the superior predictive ability of the proposed reranking model for tail labels compared to a host of competitive baselines.


Comments:
==========
The paper solves an important problem which has several industrial applications of extreme multi-label learning. The proposed methods are novel, perhaps less so to someone who is an expert in XML. The experimental evaluation is highly impressive. Both the proposed methods outperform a host of highly competitive baselines on a variety of datasets by significant margins. However, I have a couple of concerns regarding the proposed methods:


1). The RANKNET method which re-ranks the XML model's predictions needs to be compared against a baseline which also performs re-ranking for an apples-to-apples comparison in Table 2. Sure, the improvements due to reranking (vs no-reranking) are impressive, but how would a simple re-ranking approach which is not population-aware perform? How is the lambda chosen? By CV? Since you can stack RankNet modules to make it deep, how many were used for results in Table 2? How sensitive are the results to the number of modules?

2). The data augmentation for the tail labels seems arbitrary. Why only Input dropout and Input swap? Also, it is unclear how one should split the data between head and tail labels? More importantly, how are the model scores for head and tail labels integrated to make a final prediction?

---

> ### Author Response · Authors · 2020-11-13
> **Response to Reviewer 2**
>
> Thank you for your comments. We address your concerns in the following point by ponit.
>
> 1. For implementations of RANKNET, we simply set $\lambda$ as 1 in the experiments and only one RankNet layer is used.
> 2. Data augmentation strategie are not the main contribution of this work and we choose two simple and effective approaches, i.e., the input dropout and input swap, otherwise the computational overhead is huge in XML. We conducted ablations to study the influence of head and tail label partitioning threshold in Figure 5 in the supplementary material. Note that the partition rule is still an open problem in XML. The prediction of head labels and tail labels is integrated by using a binary classifier which produces probabilities of an instance related to head labels and tail labels.

---

> > ### Comment · AnonReviewer2 · 2020-11-20
> > **Thank you.**
> >
> > I have read the reviewers' response. They answered some of my questions, but I still have concerns regarding the novelty of the paper and also the lack of comparison against baselines that perform re-ranking. So, my evaluation stays the same. I think it is a good paper, but a little below the bar for ICLR.

---

### Official Review · AnonReviewer1 · 2020-10-28
**output post-processing for tail-label detection in extreme multi-label learning**

**Rating:** 4
**Confidence:** 4

**Review:**

The paper presents a method for improving tail-label performance in extreme multi-label learning setup where the number of target labels can be extremely large. It is based on the finding that the distribution of the norms of the learnt weight vectors also follows a power-law as does the distribution of the samples among labels. The main contribution of the paper is proposing methods for re-ranking which encourages precedence of tail-labels and a data augmentation mechanism. It achieves improvements when applied to sota methods on relevant PSP metrics.

Some of the concerns regarding the paper are :
- The approach overall seems more like an ad-hoc post-processing step rather than a learning algorithm. It is possible that the impact of RankNet proposed in section 3.2 can be achieved in a more simple way of reranking scores. In the code provided, it was not clear where RankNet as described in section 3.2 was implemented.
- The theorem 1 seems incorrect. The probability model is not completely specified as it is not clear what exactly is meant by the test point being randomly sampled. Is it uniformly at random (as seems to be from the proof) or from the distribution that is same as the training distribution (as the typical i.i.d assumption in ML). Also, it seems to compute expectation of some event {y_j \in \beta^{k}}, which is strange as expectations can be computed only of random variables. Overall, the statement of the theorem seems quite vague and imprecise. There are some notational issues also, the W, and w symbols in the theorem don't match the preceding text.
- In terms of the experimental results, it is not clear what happens with vanilla p@k and nDCG@k. Even though it is mentioned on page 6 para2 that the these metrics are computed but these are not given anywhere. Also, the Table 4 does not seem to be of much consequence as the re-ranking method can be potentially be applied to all the competing methods.
- Other minor comments - the references are improperly given. In some places abbreviations are used for conference names, and in others full names are given. In many places, arxiv versions of the papers are mentioned, even though the corresponding papers are published with conferences/journals.

---

> ### Author Response · Authors · 2020-11-13
> **Response to Reviewer 1**
>
> Thanks for your comments. We address your concerns in the following point by ponit.
>
> 1. The reranking module can either be learned from data or by using pre-calculated label propensities. This step can be applied to many extant XML models to facillitate the prediction of tail labels.
>
> 2. In the updated version, we explicitly specify that testing points are uniformly sampled from the same distribution as training data, and the expectation is taken over testing point x.
>
> 3. We report the results w.r.t. vanilla p@k and nDCG@k in Table 5 in the supplementary material.
>
> 4. We have carefully checked the references and corrected improper parts.

---

### Comment · ~Tong_Wei1 · 2021-08-07
**Test**

<pre>
-----------------------------------------------------------------------------------------
	           hid_dim=2048             |              hid_dim=512
 Layers    ------------------------------------------------------------------------------
           Many  Medium   Few      All  |        Many  Medium   Few     All
 ----------------------------------------------------------------------------------------
 1       61.7      45.9        26.8    49.4
 2       60.8      44.4        24.5    48.0      59.9      44.3        25.1   47.7
 3       60.3      44.3        23.7    47.7      59.3      43.7        23.9   47.0
 -----------------------------------------------------------------------------------------
</pre>

---

### Decision · Program_Chairs · 2021-01-07
**Final Decision**

**Decision:**

Reject

**Comment:**

The paper presents some interesting insights, but all reviewers have agreed that it does not meet the bar of ICLR. The theoretical results require revision as several issues have been indicated in the reviews. The authors have tried to correct them during the rebuttal, but the reviewers remain unconvinced.  Also the novelty is limited as re-ranking is a well-known concept and decoupling of head and tail labels is an approach often used in practice across many applications.

The authors should also clarify the way the RankNet method is used and implemented to clarify the issue raised by Reviewer 1. Finally, let me notice that adjusting thresholds for labels has been considered in the XMLC literature, in the context of optimization of the macro F-measure (Extreme F-measure Maximization using Sparse Probability Estimates, ICML 2016).